# Effect of Micro-Dimple Geometry on the Tribological Characteristics of Textured Surfaces

Saood Ali [1,*], Rendi Kurniawan [1], Moran Xu [1], Farooq Ahmed [2], Mohd Danish [3] and Kubilay Aslantas [4,*]

1   School of Mechanical Engineering, Yeungnam University, 280 Daehak-Ro,
    Gyeongsan-si 38541, Republic of Korea
2   Mechanical Engineering Department, College of Engineering, Dhofar University, Salalah 211, Oman
3   Department of Mechanical and Materials Engineering, University of Jeddah, Jeddah 21589, Saudi Arabia
4   Department of Mechanical Engineering, Faculty of Technology, Afyon Kocatepe University,
    Afyon 03200, Turkey
*   Correspondence: saoodali@ynu.ac.kr (S.A.); aslantas@aku.edu.tr (K.A.)

**Abstract:** The introduction of external surface features on mating contact surfaces is an effective method to reduce friction and wear between the contact surfaces. The tribological properties of the contact surfaces can be improved by controlling the geometrical parameters (shape, size, depth) of the surface texture effectively. In the present study, the tribological properties of Al6061-T6 cylindrical workpieces with various micro-dimple-texture geometries and an AISI 52100 steel stationery block are tested experimentally, in a rotating cylinder-on-pin configuration of the friction test. The dual-frequency surface texturing method is employed to create micro-dimple textures using a polycrystalline diamond tool. The effect of a hierarchical micro-dimple texture is then investigated under boundary lubrication conditions. Hierarchical micro-dimples, with an increase in length, show a lower friction coefficient under high load and sliding speed conditions. Secondary hierarchical nano-structures help in improving the tribological characteristics by generating an additional hydrodynamic lift effect.

**Keywords:** surface texture; friction coefficient; specific wear rate; cylinder-on-pin

## 1. Introduction

A poor lubrication condition of a key component in a system can severely restrict the performance of a whole machine or a system, as the poor lubrication leads to a severe rise in friction and wear of the contact surfaces. Thus, a problem of reducing friction and wear between the contact surfaces has become a key research topic in recent times. Apart from the design of friction systems, the modification of surface topography, has become one of the popular strategies in recent times to reduce friction and wear between the contact surfaces [1–3]. Etsion and co-workers [4], first, highlighted the response of surface textures on the tribological performance of seals and piston rings in various lubrication environments. Afterwards, a number of articles by numerous researchers were published regarding the consequences of surface texture features on the tribological characteristics [5–8]. However, interestingly, all these works focus on single-scale surface textures, while, in nature we often find the existence of multi-scale surface textures simultaneously in order to achieve some specific surface properties such as the super-hydrophobicity and self-cleaning ability of lotus leaves [9] and reduction of drag force by multi-scale textures of shark and dolphin skin [4].

In recent times, a number of researchers have focused their attention on the generation of multi-scale or hierarchical surface textures on the different type of surfaces using various surface texture generation methods. Initially, laser surface texturing [10], which is the most widely used method for surface texturing, was employed for generating hierarchical surface textures. However, the requirement of a controlled environment, high-priced equipment,

and most importantly the adjuvant damage to the surrounding material, which limits the size of hierarchical structures, forced the researchers to use contact-type machining methods for hierarchical surface texture generation [11].

The vibration-assisted turning (VAT) method, in which vibration is applied to conventional turning, has successfully been used in recent times for hierarchical surface texturing. It is a single-step operation, due to which the operating cost and time reduces many folds while the efficiency of the texturing operation increases [12].

Firstly, Zhou et al. [13] developed hierarchical surface structures by engaging a non-resonant two-dimensional (2D) vibration transducer along with a fast tool servo method. Yuan et al. [14] used a vibration cutting system with the double-frequency method to create hierarchical surface textures on a flat steel specimen. They employed a 2D resonant vibration transducer and a non-resonant one-dimensional compliant vibrator in conjunction. Inspired by this, Guo et al. [15] developed a 2D-elliptical vibration texturing method for hierarchical surface texturing on cylindrical workpieces. Kurniawan et al. developed a two-frequency method for surface texturing, which shows improved surface roughness as compared to the conventional surface texturing method [16–18].

As summarized above, most of the previous studies on the generation of hierarchical surface texturing have been focused on the development of a texturing method and optimizing the machining parameters for creating the optimized hierarchical surface textures in terms of surface texture shape, size, density, and arrangement [19–24]. However, despite the unique characteristics of hierarchical surface textures, there is a wide gap in the literature concerning the effect of hierarchical surface textures and their different parameters on the friction performance under various lubrication conditions.

The main purpose of the present work is to study the influence of hierarchical micro-structures on the tribological characteristics of micro-dimple-textured cylindrical workpieces. Experiments were performed to investigate the effect of various geometric parameters of micro-dimple texture on the friction properties under pin-on-disk rotating configuration and compared with that of untextured specimens.

## 2. Materials and Methods

### 2.1. Dual-Frequency Surface Texturing

As compared to non-contact surface texturing methods (such as laser surface texturing [25]), micromachining offered the benefits of high preparation efficiency and unlimited processing materials [26], which greatly improves the surface dimensional accuracy. It is a cheap and convenient surface texturing technology as it does not require any specific tool or controlled environment [27]. The authors have developed a low-cost and convenient dual-frequency method for surface texturing in the form of hierarchical micro-dimples, which has been explained in their previous publication [17]. The principle of the dual-frequency surface texturing method is shown in Figure 1. The hierarchical micro-dimple-textured surfaces were prepared using a polycrystalline diamond (PCD) cutting tool having a nose radius of 400 µm and a rake angle of 7°.

### 2.2. Workpiece Specimen

Al6061 alloy material cylindrical workpieces, with a diameter of 28 mm and surface roughness Ra ≈ 0.5 µm, were used as untextured specimens in the present study. The material composition of the specimen workpiece is presented in Table 1. As Al6061 aluminium alloy has low density, good formability, higher specific strength, and excellent strength-to-weight ratio [28,29], they are found to be excessively used in production of aviation structures (wings and fuselages) and in the automobile industry (chassis, pistons, and crankshafts) [30]. However, the Al6061 alloy has poor tribological characteristics in boundary lubrication environments, which limits its application [31]. In this study, hierarchical micro-structures inside the micro-dimples were created on the specimen surface using a dual-frequency texturing method.

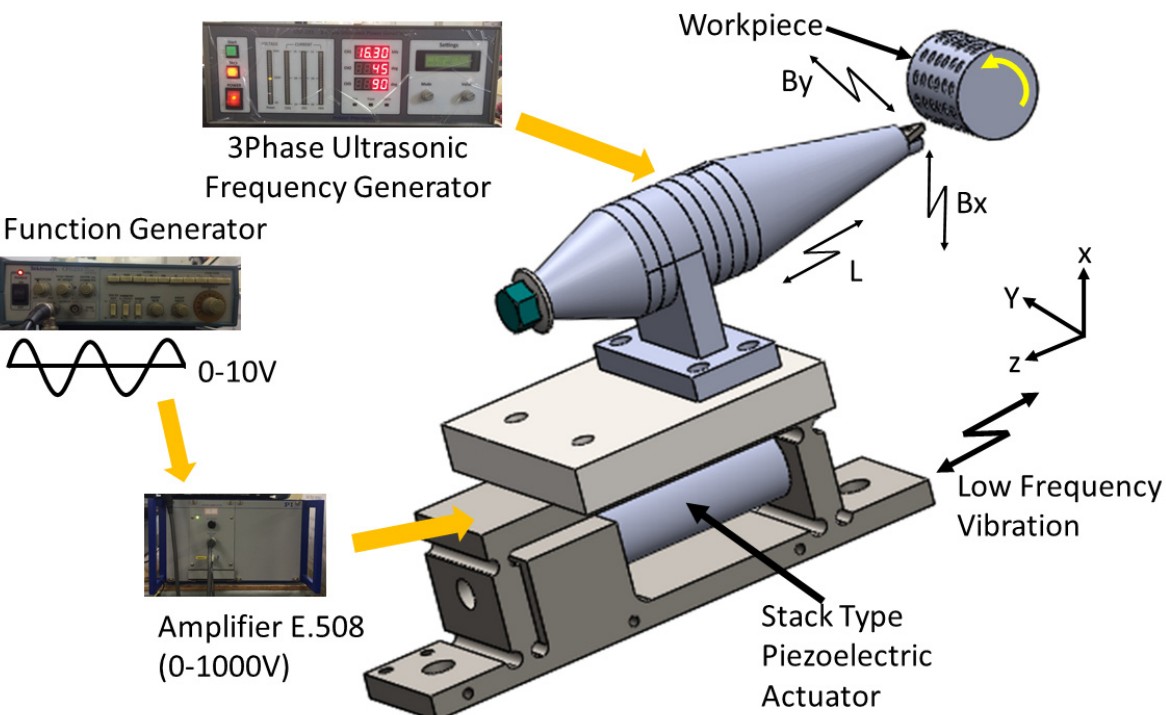

**Figure 1.** Dual-frequency resonant vibration texturing method.

**Table 1.** Al6061-T6 alloy chemical composition.

| Component | Cr | Cu | Mg | Si | Mn | Ti | Zn | Fe | Al |
|---|---|---|---|---|---|---|---|---|---|
| Chemical Composition (%) | 0.04–0.35 | 0.15–0.4 | 0.8–1.2 | 0.4–0.8 | 0.15 max. | 0.15 max. | 0.25 max. | 0.7 max. | Balance |

*2.3. Surface Texturing Operation*

The experimental setup was arranged on a CNC lathe machine (Daegu Heavy Inc. Co., Ltd., Taegu, Korea) having a resolution of 1 μm, as shown in Figure 2. The whole experimental setup consisted of a single-excitation 3D resonant vibration transducer and a low-frequency, high-amplitude displacement amplifier. The vibration characteristics of the texturing operation are shown in Table 2. The selection of machining parameters has been explained in the authors' previous publications [17,32]. The machining parameters of the surface texturing operation are given in Table 3.

**Table 2.** Vibrational characteristics for dual-frequency texturing.

| | | |
|---|---|---|
| Low vibration frequency (fl) | 230 Hz | |
| Low vibration frequency amplitude | 22.5 μm | |
| 3D resonant vibration frequency (fh) | 16.3 kHz | |
| Phase shift at 3D resonant vibration (L, Bx, By) | 0°, 45°, 90° | |
| 3D elliptical locus amplitude | Longitudinal | 1.5 μm |
| | Bending-X | 2.7 μm |
| | Bending-Y | 5.2 μm |

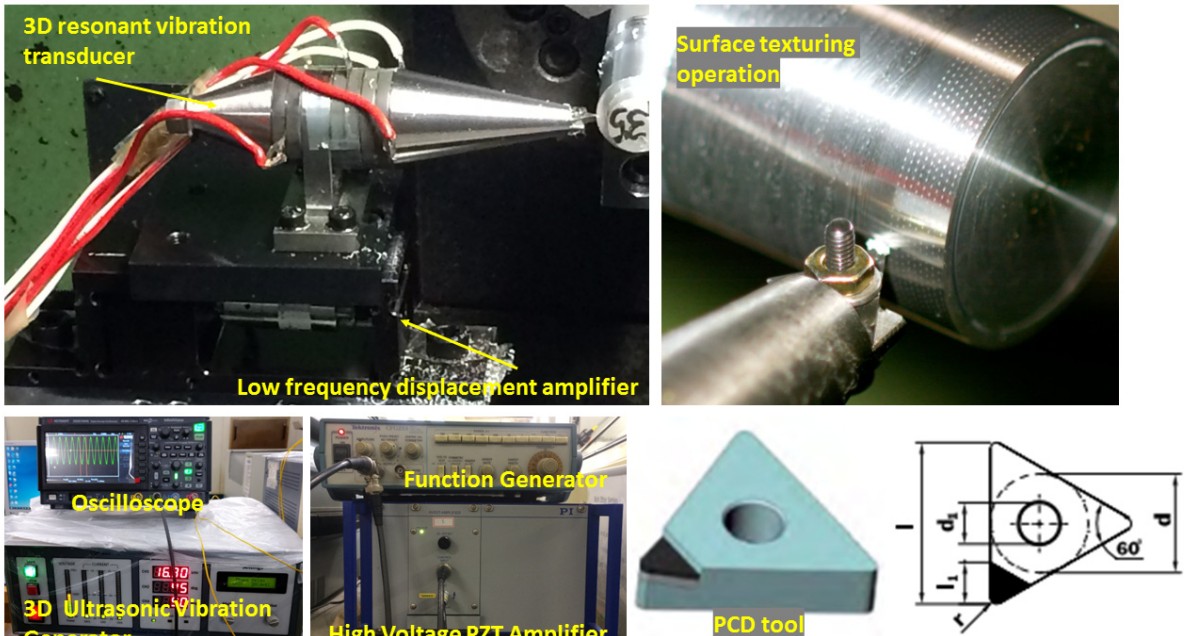

**Figure 2.** Dual-frequency texturing method—experimental setup.

**Table 3.** Experimental parameters.

| Specimen material | Al6061 alloy | | |
|---|---|---|---|
| Specimen diameter | 28 mm | | |
| Tool type | Polycrystalline diamond (PCD) | | |
| Surface machining of non-textured specimen | Spindle speed | 450 rpm | |
| | Feed rate | 0.001 mm/rev | |
| Non-textured specimen surface roughness | Ra ≈ 0.50 µm | | |
| Machining parameters of surface texturing operation | Feed rate | 1.0 mm/rev | |
| | Spindle speed (rpm) | Case 1 | 60 |
| | | Case 2 | 80 |
| | | Case 3 | 100 |
| | | Case 4 | 120 |
| | | Case 5 | 140 |

## 2.4. Tribological Experiment

The tribological properties of the textured surfaces were studied under a starved lubrication environment, i.e., near-dry lubrication. Commercially available lubrication oil that had a viscosity of ISO VG68 grade (dynamic viscosity of 0.15804 Pa·s at 25 °C room temperature) was used as the lubricant. Before the experiment, the samples were cleaned in acetone and then dried.

The friction test on the cylindrical workpieces was performed in a cylinder-on-pin configuration, as shown in Figure 3, on a commercial tribometer. The coefficient of friction was computed in real time during the experiment by measuring the normal load and friction force between the sliding surfaces. The sliding distance remains constant during the friction while the time varies in correspondence to the sliding speed. The friction characteristics were analysed under various sliding speeds with the help of 'Stribeck' curves. As the sliding speed in the block-on-ring configuration depends on the rotating speed of the cylindrical specimen, the rotational speed of the specimen was varied to achieve the different sliding speeds. The friction test conditions are explained in Table 4.

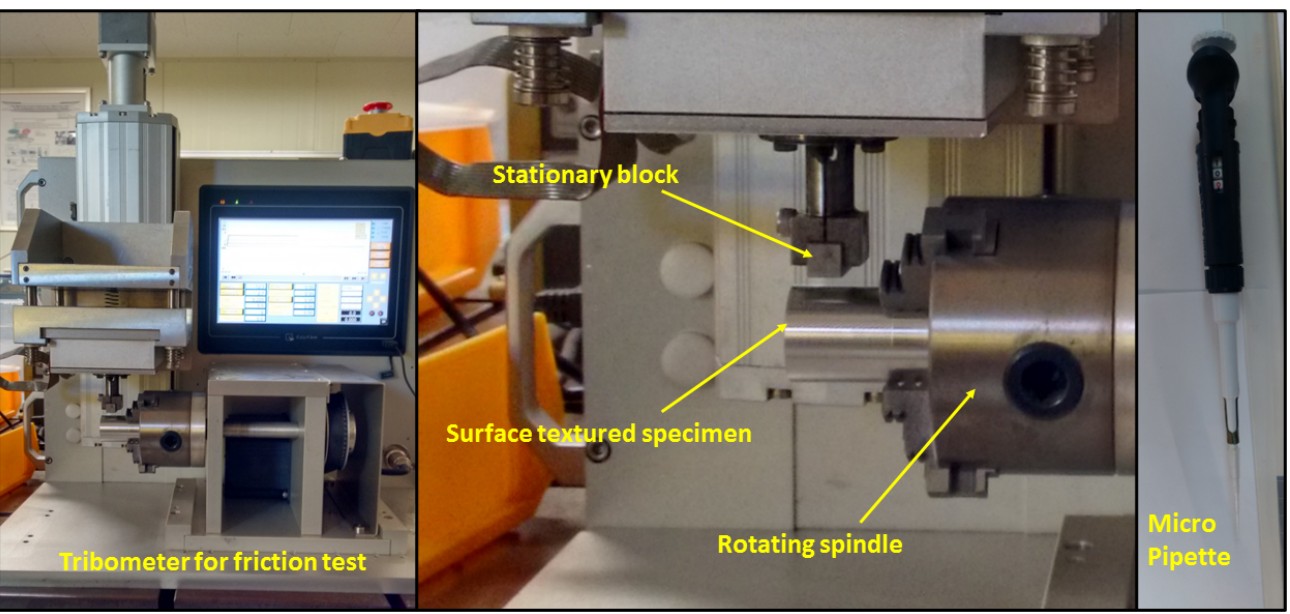

**Figure 3.** Commercial tribometer for friction test.

**Table 4.** Friction test parameters.

| Lubrication environment | Starved | |
|---|---|---|
| Load applied | 10 N, 50 N | |
| Lubricant | Commercially available lubrication oil | |
| Hertz contact pressure | 10 N | 31 MPa |
| | 50 N | 87 MPa |
| Rotating speed | 100 rpm to 1000 rpm | |

## 3. Results

### 3.1. Micro-Dimple Geometrical Parameters

Micro-dimples with and without hierarchical micro-structures on a cylindrical specimen were generated via the dual-frequency surface texturing method. Variation of the spindle speed during the texturing operation results in the generation of micro-dimples with different geometric parameters. The five different types of micro-dimples along with their profile along the length and width are shown in Figure 4. The spindle speed was chosen so that the requirement of intermittent cutting should always be met during the texturing operation [33].

The surface texture density is solely dependent on the feed rate value, as is the axial distance between any two rows of micro-dimples that follow one another [32]. For a specific federate value, the surface texture density is displayed in Figure 5. The surface texture density was determined using Equation (1) as the foundation. The textured surface profiles with various micro-dimple geometries are displayed in Figure 6. The geometric parameters of different shapes of micro-dimples are summarized in Figure 7.

$$S_d(\%) = \frac{\text{total no. of dimples in one rev.} \times \text{area of one dimple}}{\text{total area covered in one revolution}} \times 100\% \qquad (1)$$

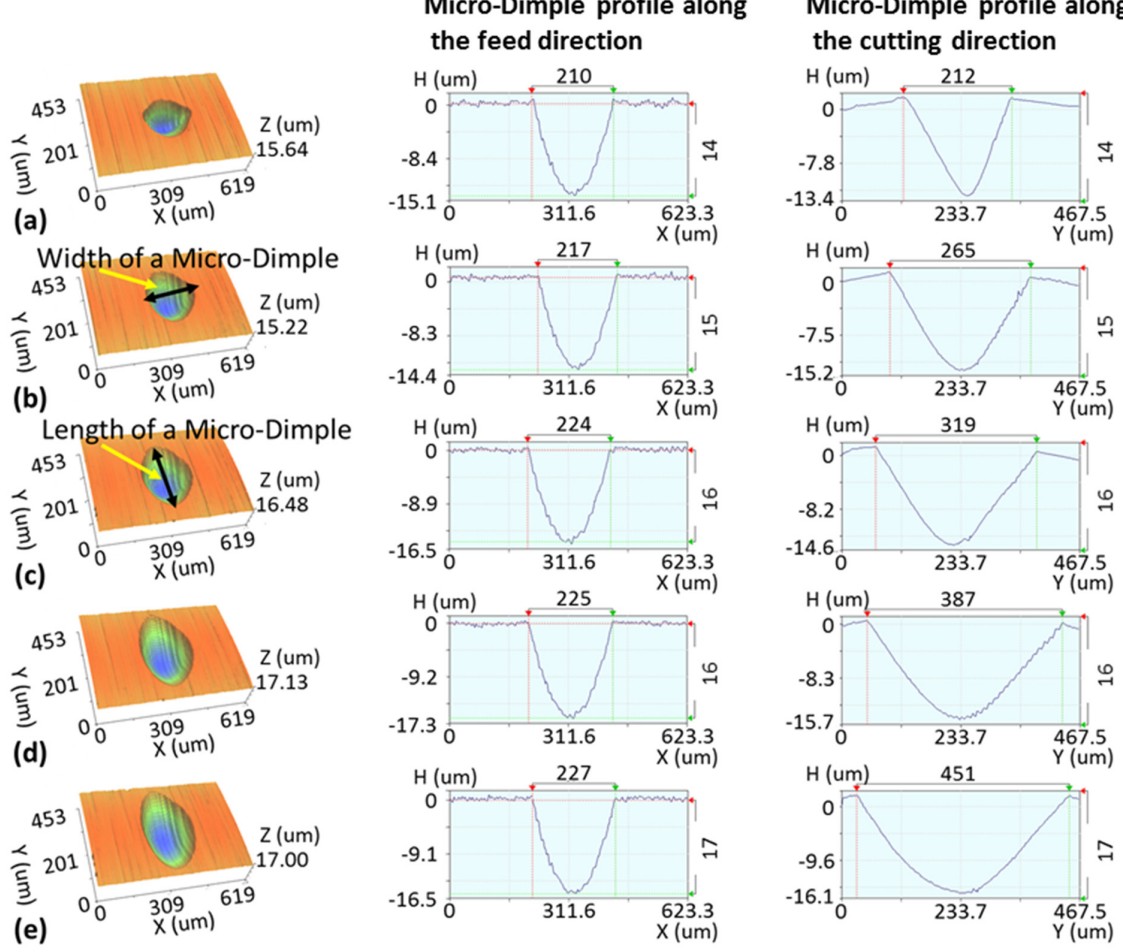

**Figure 4.** Micro−dimple geometrical profiles along the feed and cutting direction at various spindle speeds: (**a**) Case 1, (**b**) Case 2, (**c**) Case 3, (**d**) Case 4, (**e**) Case 5.

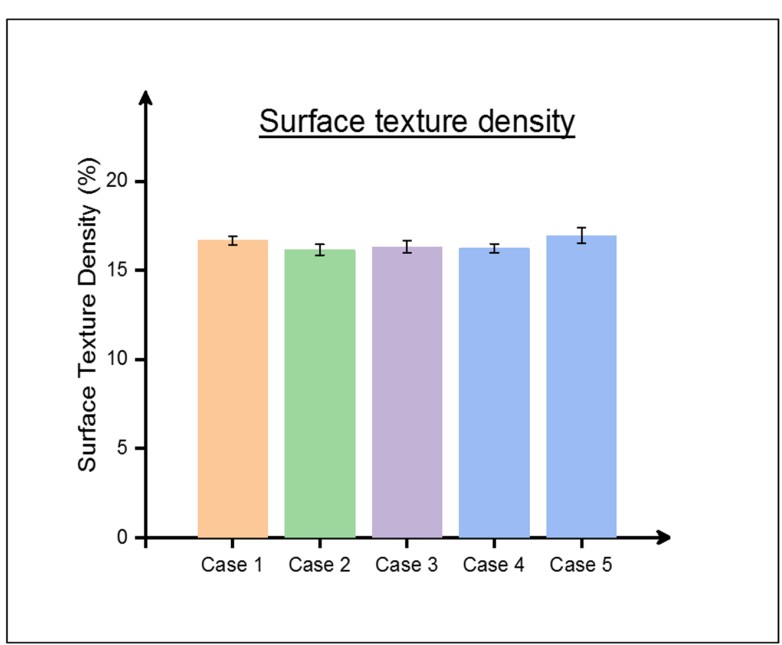

**Figure 5.** Surface texture density for different micro-dimple shapes.

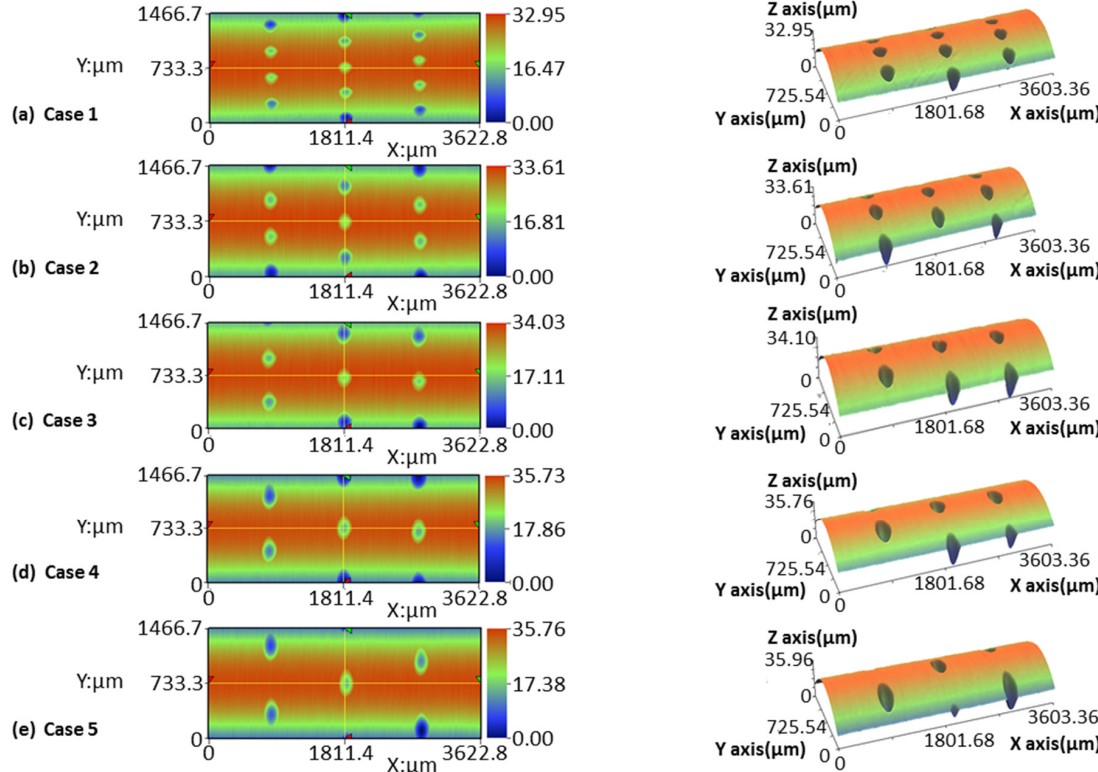

**Figure 6.** Micro-dimple-textured surface profile at a feed rate of 1.0 mm/rev with different micro-dimple shapes: (**a**) Case 1, (**b**) Case 2, (**c**) Case 3, (**d**) Case 4, (**e**) Case 5.

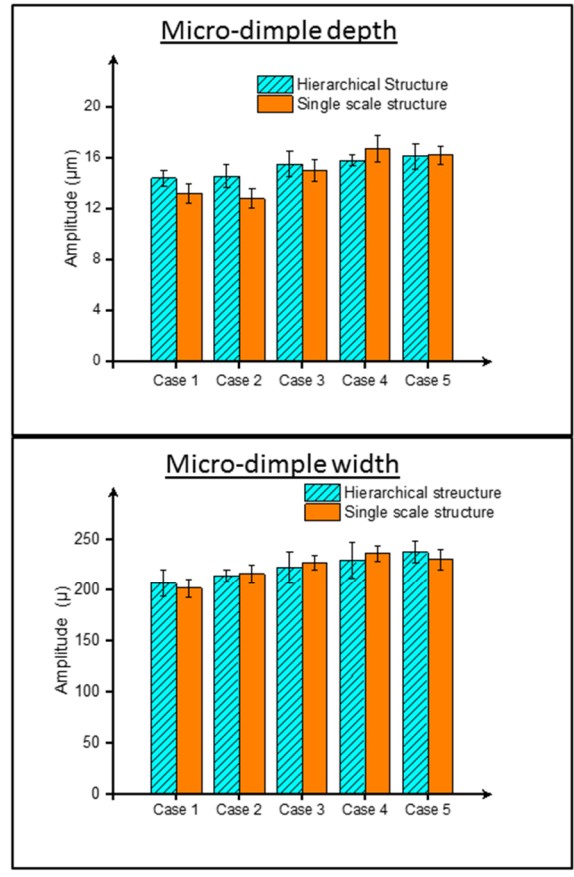

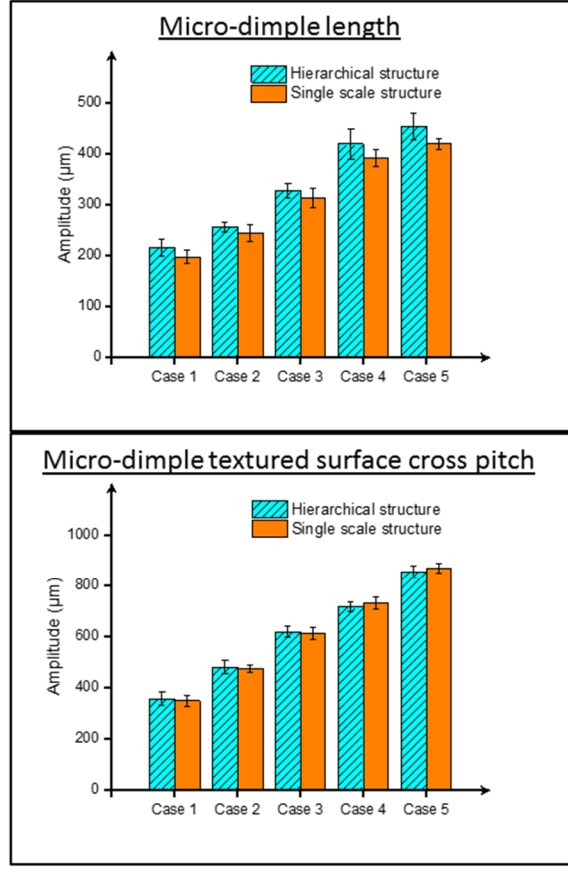

**Figure 7.** Micro-dimple-textured surface geometric parameters.

### 3.2. Effect of Applied Load on Friction Properties

The effect of applied load, i.e., the contact pressure, on the textured surfaces' friction properties is shown in Figures 8 and 9, respectively. The fluctuation of the friction coefficient with respect to the sliding speed in terms of the rotation speed of the specimen during the friction test has been reported. It was noted that the non-textured surface had a lower friction coefficient than the textured surfaces at the modest applied load of 10 N (contact pressure 36 MPa) (Figure 8). However, at a high applied load of 50 N (contact pressure 87 MPa), textured surfaces showed a lower coefficient of friction than the non-textured specimen (Figure 9). This occurs because the increase in applied load causes the lubricant to flow from the micro-dimple textures into the contact region, creating a hydrodynamic lift effect in the micro-dimples. Moreover, the micro-dimples with hierarchical structures have a lower friction coefficient than the single-scale surface textures. This is due to the lower chance of cavitation formation with hierarchical structures in a micro-dimple, which in turn increases the hydrodynamic lift effect.

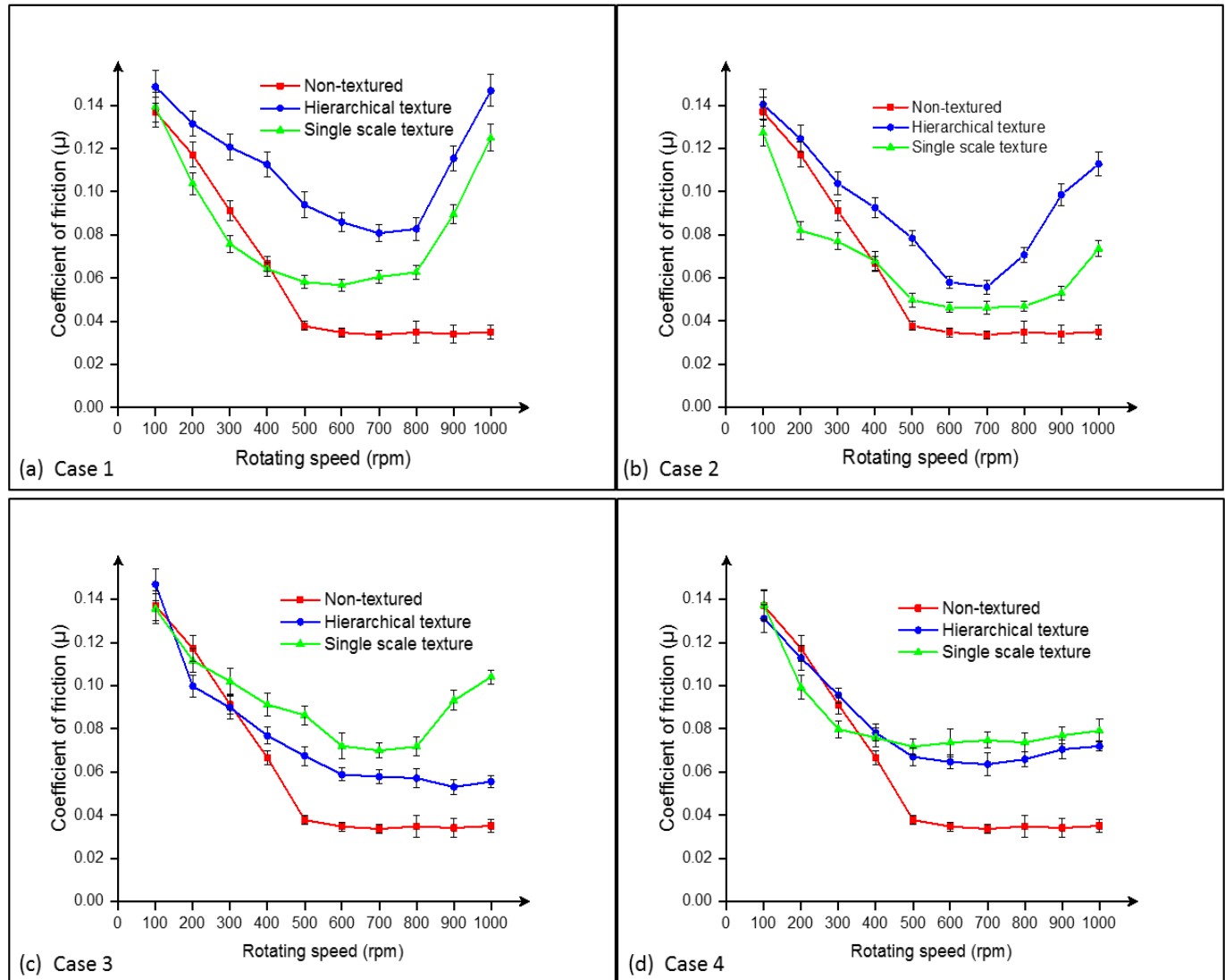

**Figure 8.** *Cont.*

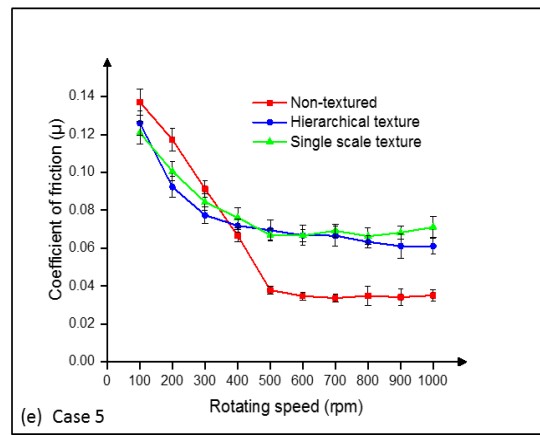

**Figure 8.** Effect of low applied load (10 N) on friction characteristics of different surfaces.

**Figure 9.** Effect of high applied load (50 N) on friction characteristics of different surfaces.

### 3.3. Influence of Micro-Dimple Geometrical Morphology on Friction Properties

The influence of micro-dimple geometrical morphology on friction properties under different applied loads is shown in Figures 10 and 11, respectively. Additionally, as shown in Figures 12 and 13, the impact of hierarchical structures on the friction characteristics for various micro-dimple geometries under various applied loads was examined.

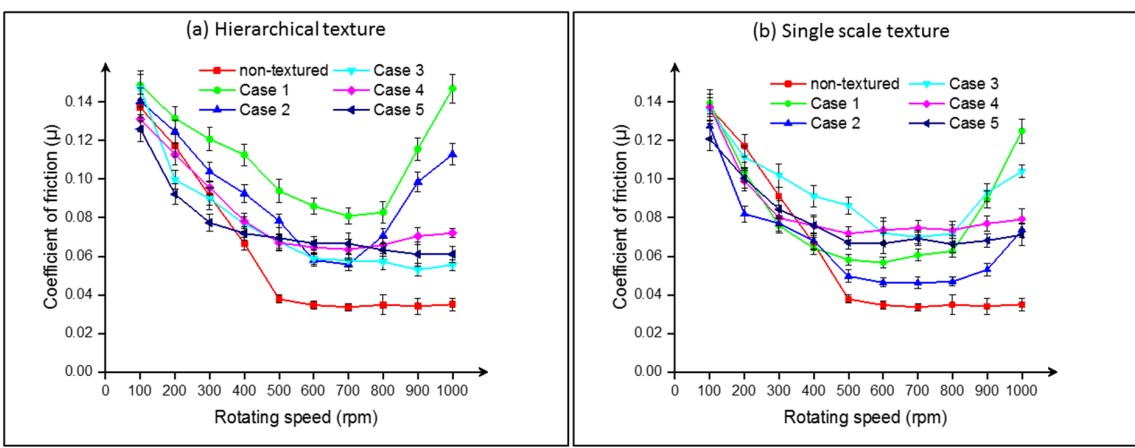

**Figure 10.** Effect of micro-dimple geometrical morphology at low applied load (10 N).

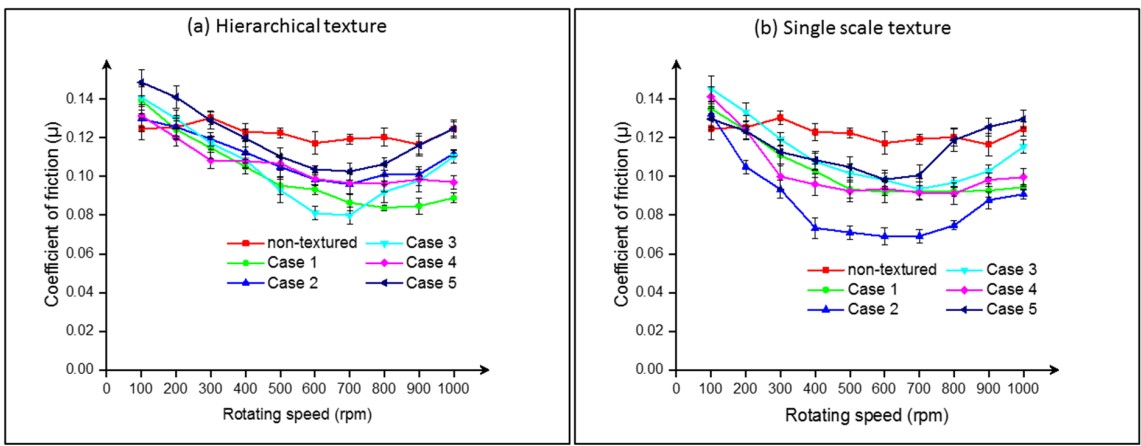

**Figure 11.** Effect of micro-dimple geometrical morphology at high applied load (50 N).

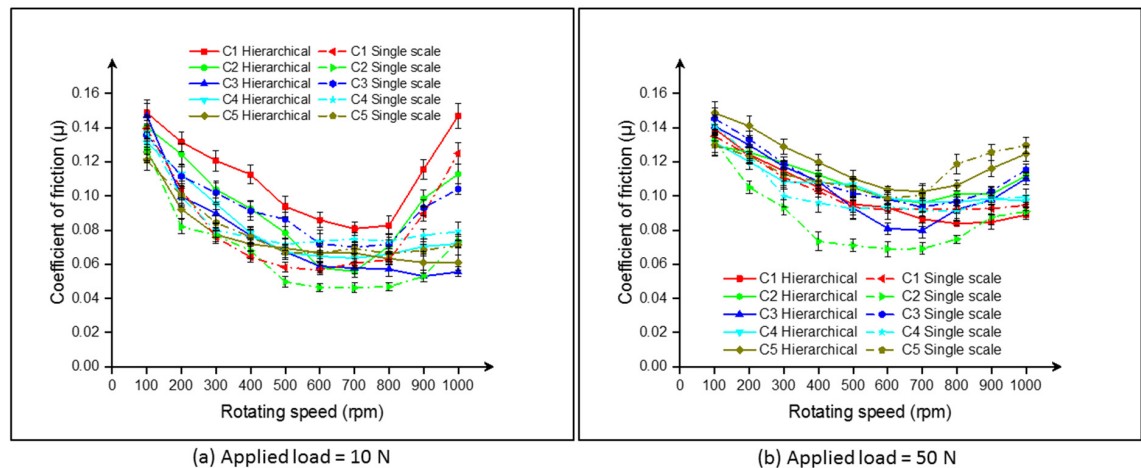

**Figure 12.** Effect of hierarchical structures as compared to single-scale structures.

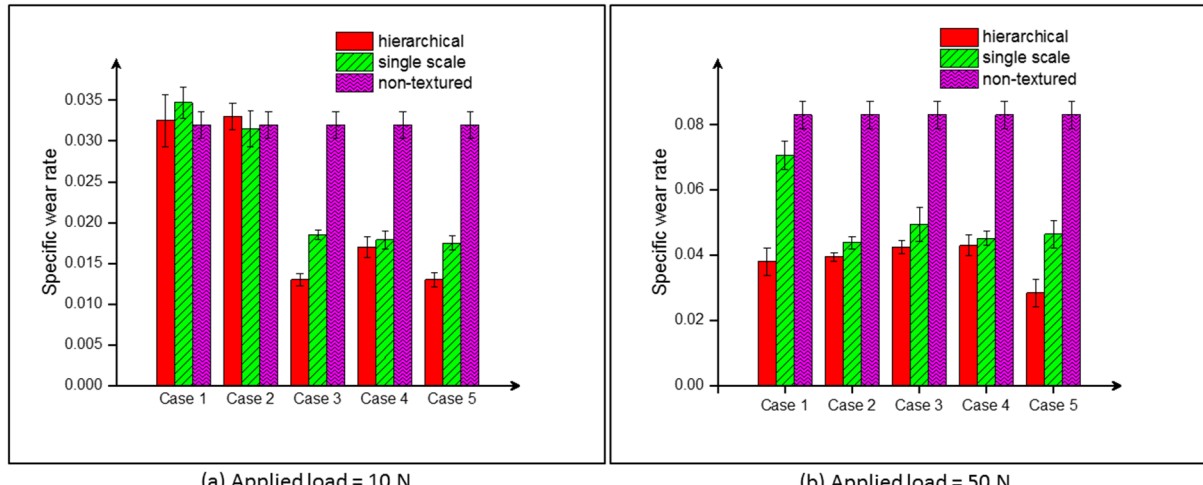

**Figure 13.** Specific wear rate of different micro-dimple shape surfaces.

It was observed that at the low applied load condition (10 N), i.e., in a low contact pressure environment, irrespective of the micro-dimple shape, non-textured specimens showed a lower coefficient of friction. Specially, in the case of high rotating speed of specimens, i.e., in the region of high sliding speed, the non-textured specimens had a decreased friction coefficient as compared to the textured surfaces. Furthermore, hierarchical structures or single-scale structure textures, both had a higher friction coefficient than that of the non-textured surface in low applied load conditions.

The textured surfaces exhibited much-improved friction characteristics than non-textured surfaces with an increase in applied load, as illustrated in Figure 11, notably in the area of high sliding speed. In high contact pressure regions, it has been shown that smaller micro-dimples offer superior friction qualities over larger micro-dimples. With an increase in micro-dimple size, in the high applied load region, there are strong chances that the whole area of the micro-dimple is not in the contact region, and therefore, the hydrodynamic lift effect is not uniform. Because of this, there is less lubricant flowing from the micro-dimples into the contact area, which increases friction.

### 3.4. Effect of Hierarchical Structures

The effect of hierarchical micro-structures as compared to single-scale structures in a micro-dimple at different applied loads is shown in Figure 12. It was observed that the single-scale structures had a lower friction coefficient than the hierarchical structures, especially with small-sized micro-dimples and in the speed range of 400 to 800 rpm. However, as the sliding speed increased in the high applied load condition, micro-dimples with hierarchical structures had a lower friction coefficient than single-scale micro-dimples. It was also observed that the small size micro-dimples with hierarchical structures were more effective in reducing the friction at high sliding speed and high contact pressure conditions. The larger micro-dimples are more effective in reducing friction under high speed and low contact pressure applications.

### 3.5. Wear Analysis

The specific wear rate ($10^{-4}$ gm/N-m) of specimens with different shapes of micro-dimples and with hierarchical structures and single-scale structures are compared in Figure 13 along with the non-textured specimen. It was observed that the small-sized micro-dimples had a higher specific wear rate at low applied load conditions as well as under high applied load. This is due to the frequent stress generation and collapse of the lubricant film around the micro-dimple boundary as the lubrication environment is starved lubrication. For the micro-dimples that are bigger in size (Case 3 onwards), the specific wear rate of the textured surfaces was much lower in comparison to that of the

non-textured surfaces. This is due to less contact between the rubbing surfaces in the case of bigger micro-dimples. Additionally, hierarchical micro-dimples had a lower specific wear rate than single-scale micro-dimples. This is because the lubricant flows more freely from hierarchical structures into the contact region than it does from single-scale structures. It is for this reason that hierarchical micro-dimples have a lower specific wear rate especially at higher applied loads compared to single-scale structures.

## 4. Discussion

The influence of hierarchical micro-structures and single-scale micro-dimple surface textures on the tribological characteristics of cylindrical workpieces under varied applied loads with a microscopic quantity of lubricant present in the contact region was studied in depth. As it is very difficult to accurately analyse the lubrication mechanism with hierarchical micro-structures, the impact of micro-dimple geometrical morphologies and contact pressure on friction characteristics was investigated. To create micro-dimples on the cylindrical workpiece, the dual-frequency resonant vibration surface texturing method was used.

The friction behaviour of the textured surfaces (hierarchical or textured surfaces) is inferior to that of the untextured surface at the low applied conditions, i.e., in a low contact pressure environment. This is because there is not enough hydrodynamic lift to allow the lubricant, which is confined in the micro-dimples, to flow into the contact area. The possibility of bulge particles becoming stuck in the micro-dimples and moving into the contact area, increasing friction, was ruled out because there were no bulges surrounding the micro-dimples, as seen in Figures 4 and 5. It was also observed that the specific wear rate of the textured surface (hierarchical/ single-scale textures) was lower than that of the non-textured surface. This was due to the lower contact area between the rubbing surfaces in the contact area. This further supports the idea that the increased friction of the textured surfaces in low contact pressure environments was primarily caused by a lack of hydrodynamic lift.

The friction properties of textured surfaces are superior to those of non-textured surfaces under conditions of high applied load (50 N). The presence of a shape effect, however, was clearly detected in the friction behaviour of the textured surfaces. In comparison to bigger micro-dimples, smaller micro-dimples showed improved friction properties especially in the region of higher sliding speed, even though the quantity of lubricant stored in bigger micro-dimples was greater. This was because the bigger micro-dimples were not completely inside the contact region and hence the hydrodynamic lift effect was not created around the whole micro-dimple cross-section resulting in a lower quantity of lubricant flowing into the contact area while this was not the case with smaller micro-dimples.

The main advantage of bigger micro-dimples with hierarchical micro-structures was observed with respect to their specific wear rate. In comparison to their similar single-scale micro-dimples, the hierarchical micro-dimples demonstrated a reduced specific wear rate although this was in contrast to their friction performance. However, compared to single-scale structures, hierarchical micro-structures were more successful in storing the worn-out particles because they limit the creation of cavitation inside a micro-dimple. This, in turn, tends to lessen the wear rate.

In summary, adding hierarchical micro-dimples to a cylindrical workpiece's surface enhances its wear and friction characteristics. Although the non-textured workpieces had lower friction characteristics in low applied load conditions, the better performance of textured surfaces in high sliding speed regions, especially in high applied load conditions, promotes the use of textured surfaces in applications where high applied load and boundary lubrication conditions are encountered frequently. Additionally, as opposed to single-scale structures, the inclusion of hierarchical micro-structures enhances the wear characteristics of the textured surfaces. It is also one of the factors that promote the use of hierarchical micro-structure-textured surfaces.

## 5. Concluding Remarks

Using a block-on-ring tribological test, the tribological properties of hierarchical micro-dimples produced by the dual-frequency resonant vibration technique on a cylindrical workpiece were investigated. The coefficient of friction and specific wear rate of different surfaces were analysed. The effect of applied load, micro-dimple geometrical morphology, sliding speed, and hierarchical structures were examined.

1.  In low applied load conditions, the non-textured surface's friction coefficient was lower than that of textured surfaces.
2.  In high applied conditions, and particularly in regions with high sliding speeds, the textured micro-dimple surfaces exhibited a lower friction coefficient than non-textured surfaces.
3.  The shape effect dominated the friction qualities of micro-dimple-textured surfaces, with smaller micro-dimples having superior friction characteristics.
4.  The inclusion of hierarchical micro-structures aided in the generation of extra hydro-dynamic lift effect, which improved the friction properties.
5.  The specific wear rate of the hierarchical micro-dimple-textured surfaces was lower than that of the comparable single-scale-textured surfaces as the hierarchical structures prevented the formation of cavitation inside a micro-dimple.

**Author Contributions:** S.A.: conceptualization, methodology, investigation, and writing—original draft preparation. R.K.: writing—reviewing and editing and investigation. X.M.: visualization, resources. F.A.: investigation and validation. M.D.: investigation and validation. K.A.: review. All authors have read and agreed to the published version of the manuscript.

**Funding:** This research received no external funding.

**Conflicts of Interest:** The authors declare that they have no known competing financial interest or personal relationships that could have appeared to influence the work reported in this paper.

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
