# Peer review of "Effect of Micro-Dimple Geometry on the Tribological Characteristics of Textured Surfaces"

_lubricants, doi:10.3390/lubricants10120328_

Round 1

Reviewer 1 Report

An interesting and well-presented paper.

Apart from a few minor English editing needs, it is well presented, but I suggest you improve it by making small changes in the presentation of the results, especially in figures 4-11, instead of labelling Case1-5, better use "Spindle speed". It would be much more appropriate and useful.

Reviewer 2 Report

The manuscript deals with the influence of hierarchical microstructures and single-scale micro-dimple surface textures on the tribological characteristics of cylindrical workpieces under 2 applied loads with a microscopic quantity of lubricant present in the contact region.

The work presented, has great value, and the information and analysis shown are quality and interesting, the only recommendation would be that include in the manuscript information about the lubricant used, at least the physical properties in order to improve the investigation.

Reviewer 3 Report

The paper reports an interesting and very useful experimental work, well structured in the manuscript. The manuscript has some weaknesses. Mentioned below aspects should be taken into consideration during the revision:

Units and abbreviations:

I suggest adding "Nomenclature" section in the manuscript;

Conclusioins:

a.            This section is well explained, but... The most important conclusions are addressed. Please, can you change the title of this section to "Concluding Remarks"?

b.            The conclusions should be in a quantified form.

c.            The practical usefulness of the results should be emphasized.

d.            The main limitations of the present method must be identified and discussed in the end of this section.
